# New Approaches to Targeted Therapy in Melanoma

**DOI:** 10.3390/cancers15123224

**Published:** 2023-06-17

**Authors:** Manuel Felipe Fernandez, Jacob Choi, Jeffrey Sosman

**Affiliations:** Robert H. Lurie Comprehensive Cancer Center, Division of Hematology/Oncology, Feinberg School of Medicine, Northwestern University, Chicago, IL 60611, USA; manuel.fernandez@northwestern.edu (M.F.F.); jacob.choi@northwestern.edu (J.C.)

**Keywords:** melanoma, targeted therapy, MAPK pathway, RAS, BRAF

## Abstract

**Simple Summary:**

Immune checkpoint inhibitors and BRAF/MEK inhibitors are the cornerstone of treatment for melanoma; however, primary and acquired resistance to these therapies highlight an ongoing, unmet need to develop novel treatment modalities. The emergence of monoclonal antibodies and small molecules as therapeutic platforms permits the targeting of specific mediators that drive the cancer phenotype. Melanoma represents a disease in which several driver mutations have been discovered, yet only a few effective targeted therapies exist. The effective targeting of these molecules may be the key to unlocking several potential novel therapies. Here we review the persisting efforts to identify and exploit molecular targets to optimize clinical outcomes for patients with melanoma.

**Abstract:**

It was just slightly more than a decade ago when metastatic melanoma carried a dismal prognosis with few, if any, effective therapies. Since then, the evolution of cancer immunotherapy has led to new and effective treatment approaches for melanoma. However, despite these advances, a sizable portion of patients with advanced melanoma have de novo or acquired resistance to immune checkpoint inhibitors. At the same time, therapies (BRAF plus MEK inhibitors) targeting the *BRAF^V600^* mutations found in 40–50% of cutaneous melanomas have also been critical for optimizing management and improving patient outcomes. Even though immunotherapy has been established as the initial therapy in most patients with cutaneous melanoma, subsequent effective therapy is limited to *BRAF^V600^* melanoma. For all other melanoma patients, driver mutations have not been effectively targeted. Numerous efforts are underway to target melanomas with NRAS mutations, NF-1 LOF mutations, and other genetic alterations leading to activation of the MAP kinase pathway. In this era of personalized medicine, we will review the current genetic landscape, molecular classifications, emerging drug targets, and the potential for combination therapies for non-*BRAF^V600^* melanoma.

## 1. Introduction

Melanocytes originate from the neural crest progenitors during embryonic development and are pigmented cells that produce melanin as a photoprotective response. Although they are most often found in the skin and hair, melanocytes are also found in the mucosal surfaces and the uveal tract of the eye. Clonal proliferation of melanocytes in these respective anatomic regions give rise to cutaneous, mucosal, and uveal melanomas.

The incidence of melanoma has been rising in the USA for the past several decades and is estimated to be 97,610, with 7990 deaths annually in 2023 [1]. The strongest risk factors for cutaneous melanomas include chronic ultraviolet (UV) exposure in older adults and blistering sunburns during adolescence and childhood; this is supported by the UV-related C > T nucleotide substitutions that predominate the genetic signature of cutaneous melanomas. At the molecular level, this malignant transition from melanocyte to melanoma is characterized and apparently dependent upon the constitutive activation of the RAS-RAF-MAPK/ERK pathway, which regulates cell proliferation, invasion, angiogenesis, and metastasis. The identification of driver mutations, such as *BRAF^v600E/K^*, along this pathway has led to the development of targeted therapy with dual BRAF and MEK inhibition, which has contributed to the remarkable improvement in overall survival of patients with *BRAF^V600^* advanced melanoma [2] (Figure 1A).

The therapeutic landscape for melanoma has come a long way since the days when the prognosis for patients with advanced, metastatic melanoma was measured in months with dacarbazine and interleukin-2 as the only FDA-approved treatment options. The emergence of immune checkpoint-based immunotherapy as the frontline treatment has significantly changed the outcome, with the most recent update from Checkmate-067 showing a median overall survival (mOS) of six years with combination ipilimumab and nivolumab [3]. Despite the unprecedented success of immune checkpoint inhibitors (ICIs), the majority of patients will eventually experience disease progression due to primary refractory disease or acquired resistance [4]. Furthermore, the decreased efficacy of ICIs in patients with other melanoma subtypes such as acral, mucosal, and uveal melanomas is reflected in poorer clinical outcomes when compared to patients with cutaneous melanomas [5,6,7,8]. Lastly, some unique populations of patients with severe autoimmune disease, organ transplants, or other medical co-morbidities who are poor candidates for immunotherapy are in need of therapeutic options. The need for alternate treatment modalities remains significant. In this review, we highlight the latest developments in targeted therapies for melanoma.

## 2. RAF Mutations, MAPK Pathway

In melanoma, molecular targets are largely within the MAPK signaling pathway, which appears critical to melanoma progression. The pathway regulates a wide range of cellular activity, including cell proliferation, survival, angiogenesis, and migration. The RAF family of proteins consists of three enzymes: ARAF, BRAF, and CRAF (RAF-1), which are activated by upstream RAS and in turn activate downstream MEK proteins via phosphorylation [9]. BRAF is dominant among the three isoforms of RAF, and the *BRAF^V600^* mutations are present in approximately 8% of human tumors, with activating somatic mutations of *BRAF* in 40–66% of melanomas [10,11]. The most frequent mutation, accounting for 90–95% of the cases, is a missense mutation causing an amino acid change from a valine (V) to glutamic acid (E) or lysine (K) at codon 600, located near the protein’s activation loop [12]. Consequently, the interaction between the activation loop and its regulatory subunit, the phosphate-binding loop, is disrupted, leading to a constitutively activated MAPK signaling cascade to drive oncogenic transformation. The established BRAF- and MEK-inhibitor combinations such as vemurafenib/cobimetinib, dabrafenib/trametinib, and encorafenib/binimetinib that target this mutation and the downstream MEK enzymes have significant clinical efficacy with increased overall and disease-free progression survival over the single agent BRAF inhibitor (dabrafenib, vemurafenib, and encorafenib) as demonstrated in the Combi-D [13,14], CoBRIM [15,16], and COLUMBUS [2,17] studies, respectively, and are utilized in clinical practice in patients with advanced/metastatic melanoma and in the adjuvant setting [18].

## 3. BRAF Mutant Resistance

Acquired resistance with clinical relapse is nearly uniform in all patients treated with the BRAF/MEK inhibitor combination. This is due in part to the numerous mechanisms of drug resistance, which make drug resistance difficult to prevent or treat effectively. These include BRAF splice variants that dimerize, BRAF amplification, NRAS mutations, mitogen-activated protein kinase 1/2 (MEK1/2) mutations reactivating ERK downstream, CRAF overexpression, and rare BRAF V600E secondary mutations such as the L505H mutation [19]. Additionally, resistance may be mediated by the overexpression or hyperactivation of membrane receptors/RTKs, the activation of parallel pathways such as aberrations in the PI3K-AKT pathway or by the direct induction of the RAS pathway, MITF copy gain, enhanced AKT signaling, down-regulation of STAG2 or STAG3 expression, suppressed CTCF-mediated expression of DUSP6, and the activation of the YAP/TAZ pathway, among other mechanisms [20].

## 4. Pan-RAF Inhibitors

Pan-RAF inhibitors have been developed as an alternative approach for preventing or overcoming resistance and the paradoxical activation of BRAF-specific inhibitors in BRAF-WT- and *NRAS*-mutated patients, as well as in the subset of *BRAF^V600E^*-mutated patients with acquired resistance. Though all RAF inhibitors are ATP-competitive kinase inhibitors, they differ in the conformation of the αC-helix and DFG motif they form. Whereas BRAF-selective inhibitors such as dabrafenib and vemurafenib bind in the DFG-in/αC-helix-out conformation [21], pan-RAF inhibitors in a DFG-out conformation achieved inhibition of BRAF and CRAF without paradoxical activation, thereby inhibiting the activity of both *BRAF^V600^*-mutant and *NRAS*-mutant disease in preclinical models [22]. Further preclinical studies undertaken with LY3009120, a pan-RAF inhibitor, demonstrated inhibition of all three RAF isoforms with similar affinity for inhibition of RAF dimers. In vivo studies using colorectal, lung, and melanoma (NRAS^Q61K^ SKMel-30) models showed dose-dependent tumor growth inhibition across *KRAS*- and *NRAS*-mutated tumors [23].

Despite this robust preclinical evidence of the activity of pan-RAF inhibitors in melanoma, several phase I trials with different pan-RAF inhibitors have seen only modest to no activity to date. A first-in-human phase I, multicenter trial of RAF265 (NCT00304525) was undertaken to establish the maximum tolerated dose and anti-tumoral efficacy of pan-RAF inhibition as monotherapy. At tolerable doses, an anti-tumor response of 12.1% was observed, irrespective of BRAF mutation status. A higher proportion of patients (20.7%) demonstrated a metabolic response and alterations in modulators of angiogenesis [24]. An additional study drug (LY3009120) was investigated in a phase 1 dose escalation/confirmation study (NCT02014116) in patients with metastatic melanoma, colorectal cancer, and non-small-cell lung cancer. The patients included in the analysis were divided into three study cohorts: (a) those with advanced melanoma with *BRAF^V60^*^0^ mutations that had relapsed after treatment with BRAF, MEK, or BRAF/MEK combination therapy; (b) advanced melanoma harboring a *NRAS^Q61X^* mutation; and (c) advanced NSCLC or colorectal carcinoma with a *KRAS* or *BRAF* mutation. Although adequate plasma concentrations associated with tumor regression in preclinical models were achieved, no partial or complete responders were seen in any group. Only 8 patients included in the trial had stable disease, with 5 of the 8 in the non-melanoma cohort [25]. Lifirafenib (BGB-283), a RAF dimer and EGFR kinase inhibitor, was assessed in a phase I trial that included patients with various advanced solid tumors. Anti-tumoral activity was observed among melanoma patients with *BRAF* mutations, with 8 of 53 (15.1%) achieving PR and 27 of 53 patients with stable disease [26]. Naporafenib (LXH254), a RAF inhibitor with potent BRAF and CRAF inhibitor activity while sparing ARAF [27], is currently undergoing investigation in a phase I trial in solid tumor patients with MAP-kinase pathway alternations (NCT02607813).

Given that the early studies showed limited clinical efficacy of pan-RAF monotherapy, mechanisms of resistance were elucidated using melanoma cells that were treated with and ultimately developed resistance to belvarafenib, a pan-RAF inhibitor. In vitro, belvarafenib-resistant clones were found to have reactivation of the MAPK pathway via mutations in ARAF leading to dimerization and kinase activity. In vivo, ctDNA obtained from patients with progressive disease following treatment with belvarafenib was also found to have ARAF mutations. These patient-derived mutations were subsequently demonstrated to confer resistance to *BRAF^V600E^* and *NRAS^Q61L^* cells in vitro [28].

Combination therapy approaches with pan-RAF have been studied to overcome acquired resistance. Preclinical data supported the use of Type II RAF and MEK inhibitors utilizing several models of MAPK-inhibition resistance in *NRAS^mut^*, *BRAF^V6000EMUT^*, and *NF1*^-/-^ cell lines [29]. This ability to overcome resistance was also seen in vivo with pan-RAF/MEK inhibitors, in part due to CD8+ TIL-mediated tumor regression with concomitant expansion of central memory T-cells and the regression of T-reg compartments. The addition of PD-L1 inhibition extended the duration of the response. Pan-RAF inhibitor therapy was also shown to sensitize *KRAS*-, *NRAS*-, or *BRAF*-mutated tumors to CDK4/6 inhibitors both in vitro and in vivo [30]. Several early-stage trials are underway investigating different pan-RAF inhibitors in combination with other agents. Currently, there are two early-phase studies exploring the combination of a pan-RAF inhibitor with an MEK inhibitor: (1) Lifirafenib (BGB-283) and mirdametinib (PD-0325901) (NCT03905148); (2) DAY101 with and without pimasertib, a MEK1/MEK2 inhibitor (NCT04985604). Additional combination approaches are being investigated with naporafenib in a phase II trial (NCT04417621) with three treatment arms in which naporafenib is taken in combination with (1) LTT462, a novel ERK inhibitor, (2) trametinib, and (3) ribociclib, a CDK4/6-inhibitor. Another active phase 1b trial (NCT04835805) combines pan-RAF (belvarafenib) and MEK inhibitor (cobimetinib) with and without nivolumab in patients with *NRAS*-mutated melanoma.

## 5. NRAS

NRAS is one of three proteins in the RAS protein family, which are in their active form when bound to GTP and in their inactive form when bound to GDP. Activated RAS protein signals via the downstream MAP kinase cascade to promote proliferative and anti-apoptotic activities. NRAS is the second most common driver mutation in cutaneous melanoma, second only to BRAF, and is found in 25–30% of cases [31]. These mutations frequently occur in exon 2, leading to an AA substitution at position 61 (glutamine for arginine, lysine, or histidine) [32]. The most common oncogenic mutations in RAS interfere with the return of RAS to an inactive GDP-bound state, which is catalyzed via GTPase activating proteins.

*NRAS*-mutated patients with advanced melanoma at the time of diagnosis have a significantly shorter overall survival (15.5 months vs. 23.5 months) [33]. Despite its prevalence and importance in melanoma, RAS has historically not proven to be amenable to direct targeting due to a lack of a readily druggable pocket. GTP-competitive drugs are limited by the extremely high affinity of RAS for GTP and high intracellular concentrations of GTP. Efforts to inhibit post-translational modification via farnesyltransferase inhibitors have fallen short [34]. RAS-targeting siRNA has been shown to sensitize melanoma cells in vitro to BRAF inhibitors [35], though it has had limited applications in vivo to date. A recent breakthrough has been the approval of a KRAS^G12C^ inhibitor, sotorasib, in NSCLC [36], yet an efficacious RAS inhibitor in melanoma remains elusive, in part due to the considerable number of downstream effectors with varying degrees of activation once the pathway is engaged [37].

Given the technical challenges of targeting RAS, attention has turned to targeting different downstream RAS effectors (Figure 1B). MEK inhibitors have been employed in RAS-mutated cancer with modest efficacy. A biomarker study of MEK inhibition with binimetinib in patients with *NRAS* mutations demonstrated consistently decreased expression of pERK [38]. This was followed by a phase II clinical trial of pimasertib versus dacarbazine (DTIC), which showed a significant improvement in progression-free survival (PFS) in patients treated with pimasertib (13 vs. 7 weeks, hazard ratio 0.59, 0.42–0.83; *p* = 0.0022). Overall response rate (ORR) was also improved with pimasertib (odds ratio 2.24, 95% CI 1.00–4.98; *p* = 0.0453); however, no OS benefit was detected (9 versus 11 months, respectively; HR 0.89, 95% CI 0.61–1.30). Furthermore, 64% of patients receiving DTIC crossed over to pimasertib, and a significant increase in serious adverse events was observed in the pimasertib group (57% vs. 20%) [39]. In an open-label, phase 3 study (NEMO), patients with advanced unresectable stage IIIc or IV disease who were previously untreated or progressed on immunotherapy received either binimetinib or DTIC [40]. A total of 402 patients were enrolled and randomized to binimetinib or DTIC, and the median follow-up was 1.7 months (IQR 1.4–4.1). Median PFS was 2.8 months (95% CI 2.8–3.6) in the binimetinib group and 1.5 months (1.5–1.7) in the DTIC group (hazard ratio 0.62 [95% CI 0.47–0.80]; one-sided *p* < 0.001). In a post-treatment analysis, patients who received previous immunotherapy, median PFS was longer for those who received binimetinib than for those who received DTIC (5.5 months [2.8–7.6] vs. 1.6 months [1.5–2.8]).

Novel MEK inhibitors are being investigated. Tunlametinib (also known as HL-085) is a novel, potent selective, oral MEK1/2 inhibitor that exhibited higher inhibitory activity against MEK1/2 than selumetinib and binimetinib, both in vitro and in vivo (unpublished data). In patients with *NRAS*-mutant melanoma, tunlametinib showed a favorable PK profile, acceptable tolerability, and encouraging anti-tumor activity at the recommended phase II dose level in a phase I study [41]. In the large phase II trial, tunlametinib, in >80 patients with *NRAS*-mutant melanoma (NCT05217303) who had previously received immunotherapy, the confirmed ORR was 39.1% (95% CI, 27.1 to 52.1) with a median DOR of 6.1 months (95% CI, 4.2 to 8.9). The study follow-up time was short at 7.9 months (range, 6.6 to 9.8). The median progression-free survival was 4.2 months (95% CI, 3.5 to 5.6), and median overall survival was immature, with a 1-year survival rate of 57.2% (95% CI, 44.7 to 67.8). Similarly, another first-in-human study evaluated FCN-159, a potent oral MEK1/2 inhibitor shown to be ten times more selective than trametinib, which demonstrated improved tolerability when compared to historical rates of adverse events of binimetinib and pimasertib and had promising anti-tumoral activity [42]. Overall, MEK inhibitor monotherapy has modest clinical efficacy with a short duration of response limited by the emergence of drug resistance secondary to the on-target reactivation of the MAPK and PI3K-AKT signaling pathways, highlighting a need to develop novel combination therapies. To that end, a recent phase Ib escalation/expansion study combined naporafenib with trametinib in patients with previously treated *NRAS*-mutated melanoma. Thirty patients were enrolled in the expansion arm of the study and demonstrated a promising clinical response with ORR, median PFS, and disease control rate (DCR) of 30%, 5.03 months, and 73.3%, respectively [43]. While every patient that received the treatment experienced a treatment-related adverse event, only 2 of the 30 patients discontinued study treatment due to severe toxicity. Efforts to decrease the skin toxicity, which occurred in 80% of patients with this regimen, are in progress.

## 6. ERK

ERK, a downstream effector of the MAPK pathway, catalyzes the phosphorylation of nuclear transcription factors once it is in its active state. ERK signaling in melanoma becomes independent of upstream feedback mechanisms because of mutations in BRAF and NRAS [9]. Furthermore, ERK signaling is sustained without hyperactivation and subsequent cell cycle arrest or death due to intrinsic feedback regulation via DUSP transcription [44]. ERK signaling is restored in patients that acquire resistance to RAF and MEK inhibition [45], generating an interest in targeting ERK directly to address resistance. Kidger et al. [46] demonstrated the preclinical activity of ERK inhibitors. A majority of ERK inhibitors are ATP-competitive, with Type I inhibitors targeting the ATP-binding site when the kinase is active in the DFG-in conformation, whereas Type II inhibitors target the ATP-binding site in the inactive DFG-out conformation. Additional kinase inhibitors are either allosteric (Type III–IV), bivalent (Type V), or covalent (Type VI) [47].

Various examples of ERK inhibitors are currently undergoing clinical trials in multiple tumor types (Figure 2D). LY3214996 [48], an ATP-competitive ERK1/2 inhibitor was studied in a phase I trial both alone and in combination with other agents in advanced malignancies, including melanoma, with results due to be reported. Ulixertinib (BVD-523) was demonstrated to have robust anti-tumor activity in the preclinical models of the *BRAF*-mutated tumor, including those that had acquired resistance to BRAF/MEK-targeted therapy [49]. A phase I dose escalation and expansion study (NCT01781429) was completed in 135 patients with advanced solid tumors. Tumor regression or stabilization was seen across dose cohorts. Among melanoma patients who had previously been treated with and progressed on BRAF and/or MEK inhibitors, 3 of 19 patients (15%) achieved a PR (including a durable response in one patient that remained on study for over 38 months), 6 patients had stable disease, and 10 patients had progressive disease. Objective responses were also observed among patients with atypical BRAF non-V600E mutations [50]. On the contrary, a phase II trial of ulixertinib in 13 patients with metastatic uveal melanoma failed to show clinical activity [51]. Dermatologic adverse events were frequently observed in patients receiving ERK inhibitors, with incidences reported as high as 79% and 76% of patients treated with ulixertinib. The most reported dermatologic adverse event included acneiform and maculopapular rashes [52]. Resistance to ERK inhibition was demonstrated to develop in *BRAF/RAS*-mutated cell lines via acquired mutations, which can inhibit binding to ERK and ERK2 amplification [53]. A key mutation was identified within the highly conserved DFG motif. Of note, these mutated cells remained sensitive to MEK inhibition, suggesting a potential benefit to ERK/MEK combination therapy [54]. To test this hypothesis, MK-8353, a selective ERK1/2 inhibitor which had limited evidence for drug efficacy as a single agent [55], is currently being investigated in combination with selumetinib (MEK1/2 inhibitor) in 30 patients (NCT03745989). Preliminary data showed significant toxicity with this regimen: dose-limiting toxicities were seen in 50% of patients receiving the highest dose and 100% of patients experienced an adverse event. The safety and preliminary efficacy of MK-8353 with pembrolizumab are being explored in patients with advanced solid tumor malignancy in a separate study (NCT02972034).

## 7. Atypical BRAF Mutations

Melanomas with an atypical *BRAF* mutation encompass a diverse genetic alteration and are often subdivided into V600 and non-V600 mutants. Each mutation is further classified into classes I, II, or III based on its signaling mechanism and kinase activity, which lead to differing degrees of molecular deregulation [56]. Rare *BRAF^V600^* mutations, which include V600R/D/M/L, are kinase-activating monomers characteristically found in older, male patients with a history of chronic sun damage. Non-V600 mutants can be categorized as class II (e.g., L597P/Q/R/S, K601E, G469R/S/A) or III (G596R, D594Y/N/G/E, D287Y) depending on the formation of dimers to activate RAF kinases or heterodimers that impair the kinase activity entirely, resulting in a paradoxical activation of ERK signaling, respectively. Typically, class II and III tumors exhibit a more aggressive clinical course and are associated with a poorer prognosis. Atypical BRAF mutations are more commonly observed in mucosal melanoma [57].

While BRAF/MEK inhibitor combination therapy is well established in patients with V600E/K mutations, the clinical data for the ~5% of melanomas with non-V600 *BRAF* mutations mainly consists of case reports given their rarity and genetic heterogeneity. Following preclinical evidence of decreased phospho-ERK signaling when treated with MEK inhibition, an initial phase I N of 1 study yielded a partial radiographic response with a PR for 24 weeks in a patient with a *BRAF* L597 mutation treated with a MEK inhibitor (TAK-733) [58]. Patients with atypical *BRAF* mutations or fusions treated with trametinib achieved an ORR of 33% (three of nine patients), with the best treatment response (87% reduction, PFS 19.2 months) occurring in a patient who harbored a class III non-V600 mutation (*BRAF* T470R) [59]. A larger retrospective cohort study assessed the clinical responses of patients with atypical *BRAF* mutations who were treated with a BRAF inhibitor, a MEK inhibitor, or a combination of BRAF and MEK inhibitors. Of the 103 patients, 58 (56%) had tumors with a rare V600 mutation, 38 (37%) with a non-V600 mutation, and seven with both V600E and a rare BRAF mutation [60]. Ninety-six patients included in the study received BRAF inhibitor, MEK inhibitor, or combination treatment. Response rate was dependent on *BRAF* genotype, with the best treatment response seen in non-E/K V600-mutated melanomas when given BRAF plus MEK inhibitors (ORR 56%, median PFS 8.0 months). Patients with non-V600 mutations achieved an ORR of 28% (5 of 18) when treated with BRAF and MEK inhibitors.

BRAF kinase fusions have also been targeted with MEK inhibitors. Clinical features of BRAF fusions include a high prevalence in younger patients and tumors with spitzoid histopathologic features. In vitro data from six melanoma cell lines with representative fusion kinases demonstrated different responses to RAF/MEK inhibition based on specific features of translocation, with translocations that yielded a higher expression level being associated with more resistance [61]. Preclinical therapeutic efficacy was demonstrated in the most resistant cell lines via the combination of a third-generation (αC-IN/DFG-OUT) RAF inhibitor with an MEK inhibitor both in vitro and in vivo [62].

A two-patient case series published in 2015 [63] reported the clinical activity of an MEK inhibitor (trametinib) in pre-treated metastatic melanoma patients with BRAF fusions. Both patients reported symptomatic improvement, and one patient demonstrated a 90% reduction in extracranial disease burden. Additionally, a case report of an advanced melanoma patient with SKAP2-BRAF fusion had a partial response to MEK inhibitor monotherapy after progressing on immunotherapy and dacarbazine, though the response was not durable and the patient’s disease ultimately progressed [64]. In an international, retrospective study, there was no anti-tumoral activity in patients harboring non-V600 mutations with BRAF inhibitors, whereas MEK inhibition with or without BRAF inhibitors demonstrated clinical activity with an ORR of 28–40%, suggesting that MEK inhibition may be the primary therapeutic agent [60]. In contrast, patients with atypical (non-E/K) V600 mutations demonstrated promising responses to BRAF inhibitor monotherapy or combined BRAF and MEK inhibitors in select phase II studies [59]. Taken collectively, it suggests that there may be a role for BRAF and MEK inhibition in patients with an activating non-V600 mutation who are either ineligible or have failed immunotherapy. Further studies are needed to better define the patients with non-V600-mutated melanoma who would derive meaningful clinical benefit from MEK inhibitors with and without BRAF inhibition.

## 8. CDK4/6 + MEK Inhibition

Fifty percent of *NRAS*-mutant melanomas have genetic aberrations in cell-cycle-associated genes providing a rationale for combining MEK inhibitors with CDK4/6 inhibitors [31]. CDK4/6 inhibitors lead to cell cycle arrest by preventing the CDK4/6-Cyclin D1 complex from phosphorylating the retinoblastoma protein (Figure 2C). A preclinical study using *NRAS*-mutant human melanoma cell lines revealed synergistic effects on both apoptosis and cell-cycle arrest with corresponding tumor regression [65]. In a phase II cohort of a clinical study with 41 patients who received binimetinib and a selective CDK4/6 inhibitor, ribociclib (LEE011), ORR, median PFS, and median OS were 19.5%, 3.7 months, and 11.3 months, respectively [66]. Patients who had a co-mutation in the D-cyclin-CDK4/6-INK4a-Rb pathway exhibited improved outcomes with an ORR of 32.5% compared to 10% in patients who did not have such alterations, suggesting a synergistic anti-tumoral effect of the combined MEK and CKD4/6 inhibition in this genetically defined population. Currently, a phase II trial evaluating the efficacy of pan-RAF inhibitor naporafenib combined with ribociclib in patients with unresectable or metastatic *NRAS*-mutated melanoma is in progress (NCT04417621).

## 9. SHP2

Src homology 2-containing protein tyrosine phosphatase 2, or SHP2, has recently been the basis for novel targeted therapy approaches. SHP2 functions as a protein tyrosine phosphatase involved in the activation of various signaling pathways, including RAS/Raf/MAPK, PI3K/AKT, JAK/STAT, and PD-1/PD-L1 [67] (Figure 2D). Consequently, modulation of SHP2 activity can impact cell survival and immune regulatory pathways, making it an appealing target for drug therapy, including for patients who have developed resistance to MAPK-targeted agents [68,69,70,71].

Historically, SHP2 has been an elusive molecular target; however, the advent of allosteric inhibitors has allowed for the development of TNO155, an orally bioavailable, first-in-class allosteric inhibitor of SHP2, to be examined in patients with advanced solid tumors, including BRAF/NRAS wild-type melanoma. Preliminary results from a first-in-human dose-escalation trial demonstrated that TNO155 inhibited SHP2 activity at well-tolerated doses [72]. This study is still actively recruiting patients, with anticipated completion in early 2024 (NCT03114319). The synergistic anti-tumoral effect of TNO155 is being explored in combination with BRAF and MEK inhibitors in other solid tumor types as well (NCT04294160).

## 10. Autophagy

Melanomas that have acquired resistance to BRAF plus MEK inhibition are often characterized by rapidly progressing disease. One of the resistance mechanisms that has been identified is autophagy, an adaptive cellular process in which abnormal proteins, damaged organelles, and pathogens are broken down via lysosomal degradation for recycling of cellular components into molecular precursors and energy to facilitate self-renewal [73,74]. Specifically, MEK1/2 inhibition increases autophagic flux by activating the LKB1-AMPK-ULK1 signaling axis in RAS-driven cancer cells. Therefore, autophagy has become a rational target to overcome BRAF/MEK resistance (Figure 2B). In the recent BAMM study, the combination of dabrafenib, trametinib, and hydroxychloroquine, a lysosomotropic agent that blocks fusion of autophagosomes to inhibit autophagy, was studied in patients with advanced, *BRAF^V600^*-mutated melanoma. This phase I/II trial demonstrated one-year PFS and RR of 48.2% and 85%, respectively; however, it failed to meet its primary endpoint of one-year PFS greater than 60%, and no significant increase in durable efficacy was observed [75]. Nonetheless, there were encouraging signals in the outcomes, including an mOS of 22.2 months with the combination therapy despite the high proportion of patients with elevated serum LDH, a larger baseline tumor size, and prior lines of treatment. A placebo-controlled, randomized study of dabrafenib and trametinib with or without hydroxychloroquine in patients with elevated serum LDH is underway to further evaluate the anti-tumoral efficacy of this regimen (NCT04527549). Novel autophagy inhibitors are also being investigated. DCC-3116, a first-in-class, selective ULK1/2 inhibitor, is being combined with trametinib or binimetinib in a phase 1/2, multicenter, open-label, first-in-human study for patients with advanced or metastatic solid tumors with mutated RAS/MAPK pathways, including melanoma (NCT04892017).

## 11. Epigenetic Targets

*EZH2* encodes for a catalytic subunit of the polycomb repressive complex 2 (PRC2), which facilitates histone H3 lysine 27 trimethylation. PRC2 plays a key role in epigenic regulation via the promotion of chromatin compaction [76,77]. EZH2 is highly expressed in human primary and metastatic melanoma and found to be required for melanogenesis, proliferation, and metastases; its expression is associated with a poorer prognosis [78]. Point mutations of *EZH2* result in a gain-of-function in 5% of melanomas and are often found in coexistence with activating *BRAF* mutations, which may indicate an oncogenic synergism. Furthermore, even in *EZH2* wild-type melanomas, the expression of EZH2 is increased in approximately 30% of BRAF*^V600E^* mutated melanomas, suggesting that there may be a greater dependency on EZH2 activity in a sizable number of patients with *BRAF^V600E^*-mutated melanomas [79]. Indeed, EZH2 was demonstrated to cooperate with a *BRAF* mutation to promote tumorigenesis, and inhibition of both BRAF and EZH2 demonstrated improved efficacy when compared to single therapy in vitro and in vivo [80] (Figure 2C). Based on these genetic and preclinical data, a phase I/II study combining tazemetostat, a reversible EZH2 inhibitor, with dabrafenib and trametinib (BRAF/MEK inhibition) in patients with BRAF V600E/K mutated melanoma who have previously progressed on prior BRAF/MEK inhibitor therapy is currently active (NCT04557956).

## 12. BAP1 and HDAC

BRCA1-associated protein 1 (*BAP1*) is a tumor suppressor gene that is frequently mutated in metastatic uveal melanomas in the late stages of melanomagenesis. The mutation results in a loss of function of the gene and drives cells to de-differentiate and acquire stem-like properties with increased capacity for self-replication [81]. Alterations in epigenetic regulations contribute to this phenotype, as evidenced by the restoration of melanocyte differentiation when uveal melanoma cells with mutated *BAP1* are treated with histone deacetylase (HDAC) inhibitors (Figure 2C). However, the clinical efficacy of HDAC inhibitors has not been established. Entinostat, a pan-HDAC inhibitor, was combined with pembrolizumab in the PEMDAC trial and demonstrated modest activity: an ORR of 14%, a median PFS of 2.1 months, and a median OS of 13.4 months [82]. Interestingly, three patients with *BAP1* wild-type uveal melanoma had disease response, suggesting that the immunomodulatory function of HDAC inhibitors may be the predominant contributing function in anti-tumoral activity.

## 13. Augmenting Immunotherapy

The advent of immunotherapy has represented the most significant advance in the care of advanced melanoma patients since the initial efforts with systemic therapy many years ago. While median OS has been reported to reach 72.1 months [3] in patients treated with the combination ipilimumab/nivolumab, this approach can be limited by significant treatment-related immune adverse events and the development of resistance [83].

Various novel approaches have been undertaken to combine immunotherapy with targeted therapy agents to overcome resistance and improve outcomes (Figure 2A). A well-established approach has been to reverse the immunosuppressive effect of pro-angiogenic factors in the tumor microenvironment using tyrosine kinase inhibitors that interfere with VEGF signaling to augment the anti-tumoral efficacy of PD-1/L1 inhibitors [84,85,86]. Lenvatinib, a multiple kinase inhibitor targeting VEGF-R1, VEGF-R2, and VEGF-R3, was added to Pembrolizumab in the phase II LEAP-004 trial of patients with advanced disease who had progression on PD-1/L1 therapy. Durable responses were reported, with a median PFS and OS of 4.2 and 14 months, respectively [87]. BiCaZO (NCT05136196)—a study combining cabozantinib, a small molecule inhibitor of tyrosine kinases VEGFR2, KIT, FLT3, RET, c-MET, and AXL with nivolumab, a standard treatment regimen already approved for advanced renal cell carcinoma—is currently recruiting advanced melanoma patients as a part of a larger immunoMATCH initiative (NCT05136196) that utilizes whole exome sequencing and gene expression profiling to enroll patients in biomarker cohorts. In a similar concept, brentuximab vedotin, an antibody drug conjugate (ADC) that targets CD30-positive regulatory T cells (T-regs), is being evaluated in combination with pembrolizumab to examine its ability to overcome a well-known mechanism of PD-1 resistance mediated by immune suppressive T-regs, which are enriched in tumors and inhibit expansion of CD8 T cells to hamper the anti-tumoral activity of immunotherapy (SGN35-033).

## 14. Conclusions

The therapeutic landscape for advanced melanoma has undergone a significant expansion in the past several decades. Despite the advances in immunotherapy and targeted therapies that have revolutionized the treatment paradigm for melanoma, there is a persistent need to develop novel therapies, particularly for patients who have progressed on ICIs or for those patients who are poor candidates for immune-based therapies. The identification of driver mutations for rational design of new targeted therapies will be critically important to continue optimizing clinical outcomes and achieving the elusive cure for melanoma.

## Figures and Tables

**Figure 1 cancers-15-03224-f001:**
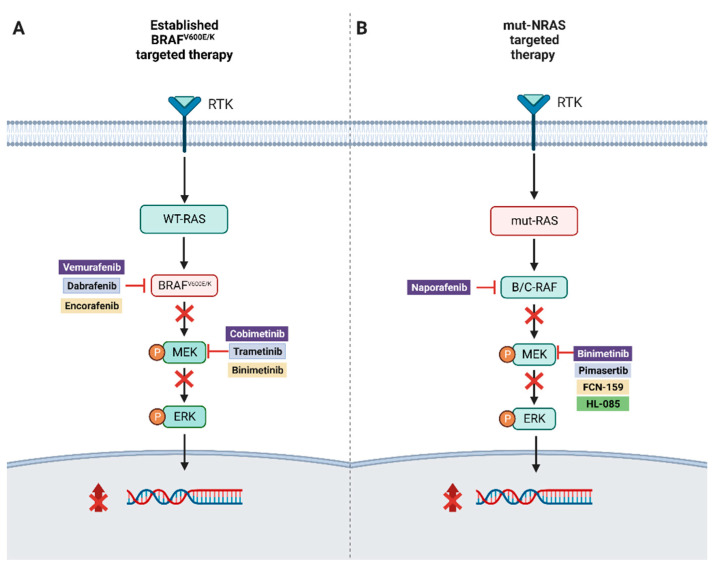
(**A**) Illustration of combination targeted therapy in *BRAF^V600E/K^*-mutated melanoma. (**B**) Targeted therapy approaches for NRAS-mutated melanoma.

**Figure 2 cancers-15-03224-f002:**
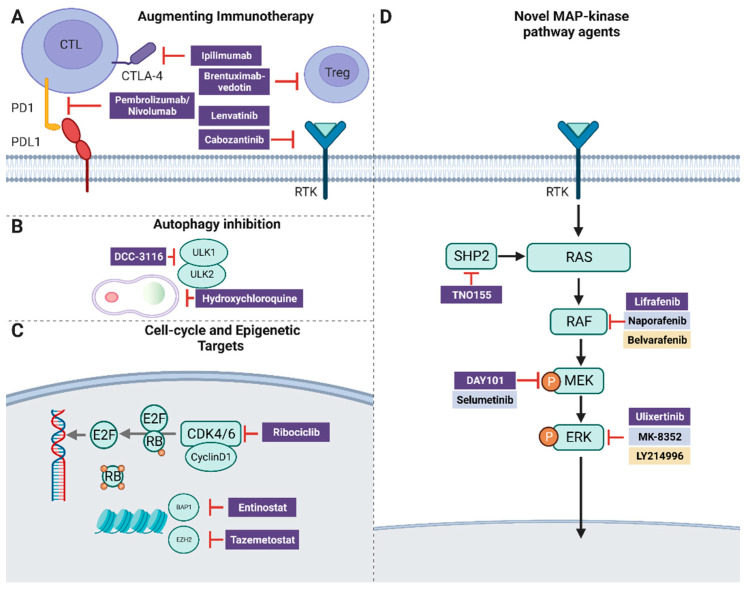
(**A**) Overcoming resistance to immunotherapy via inhibition of VEGF signaling and T-reg depletion. (**B**) Targeting autophagy with hydroxychloroquine and a novel agent. (**C**) Promoting cell cycle arrest and transcriptional regulation via small molecule inhibitors (**D**) Novel agents targeting effectors along the MAPK cascade.

## Data Availability

Not applicable.

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
