# Peer review of "New Approaches to Targeted Therapy in Melanoma"

_cancers, 2023, doi:10.3390/cancers15123224_

Round 1

Reviewer 1 Report

Dear Authors,

This is very interesting work. It does require high level of expertise regarding the molecular biology of malignant melanoma, thus my understanding may be limited for some parts of the article.

My only question is: even though it is not possible to detail all the targeted therapies, I wonder why didn'y you discuss the therapeutic interest of lenvatinib?

Best regards

F.Brunet

Author Response

Dear Dr. Brunet 

Thank you for your comments on our paper. We have added a reference to Lenvatinib and a relevant recent trial to our augmenting immunotherapy section. 

Reviewer 2 Report

This manuscript is a timely review on the different approaches to targeted therapy in melanoma. This review summarizes the importance of the MAPK signaling pathway in melanoma progression and targeted therapy. The authors describe the different ways to target the MAPK pathway highlighting the clinical studies in progress. The various emerging drug targets, and the potential for combination therapies for non-BRAFV600 melanoma are fully summarized. Moreover, the author discussed the ways to augment immunotherapy. 

The subject is important, interesting and well presented by experts in the field. This topic is of interest to the readership of Cancers. Considering all the above achievements this review has reached, this article is suitable for publication with minor modifications. 

The authors should indicate that CRAF is also called RAF-1 which is its official name. The paragraph about atypical BRAF mutations could be completed by indicating that atypical BRAF mutations are more frequent in mucosal melanoma.

Author Response

Thank you for your kind comments and your review of our paper. We made sure to denote that CRAF is also called RAF-1 by our first mention of CRAF. We also added a reference indicating that atypical BRAF mutations are more common in mucosal melanoma as you suggested. 

Reviewer 3 Report

      In this research, the authors reviewed the current development of new approaches to targeted therapy in melanoma. Generally, it’s meaningful and interesting review. In my opinion, the current version of this manuscript fits the scope of Cancers and could be accepted after major revision.

My specific comments are in detail listed below:

1.     If possible, how immune pathway regulation affect melanoma therapy could be added including PD-L1, CTLA-4, and et al..

2.     In this review, how was the current development of targeting PD-L1 pathway to better cure melanoma. Some references should be added to this part including 10.1016/j.carbpol.2021.118869.

3.     In Figure 1, some immune regulation agents used for melanoma therapy could be added.

4.     Some minor mistakes exist in the references, such as ref. 1, 7, and 36. The authors should correct it.

5.     In the review, the possibility of using mitochondria function inhibition to sensitize melanoma therapy could be added. Some references should be added to this part including 10.1016/j.ijbiomac.2022.10.167.

6.     A more depth outlook or prospect that pointing out the future clinical therapy direction of melanoma treatment could be added.

Author Response

Thank you for your thoughtful review our paper. 

1, 2) We agree that the development of immunotherapy has represented an important advance in melanoma treatment, however, we chose to narrow the scope of our review to focus on molecular targeted therapy and only included a brief section on how targeted therapy approaches can augment current immunotherapy. 

3) We have modified our figure to include a separate panel illustrating our "augmenting immunotherapy" section. 

4) We revised the references you mentioned.  

5) While the possibility of utilizing chitosan to enhance PDL-1 therapy and sensitize cells to traditional chemotherapy is an exciting area of inquiry, we only included therapeutic approaches in our review that have been or are currently being evaluated in clinical trials. 

6) We included future directions of treatment within each of the subsections that we outlined and have added an additional reference to an open clinical trial in overcoming resistance to PD-1 therapy. 

Round 2

Reviewer 3 Report

The current version of this manuscript could be accepted